# Selective activation of ipRGC modulates working memory performance

Yuta Suzuki[1]*, Shigeki Nakauchi[2], Hsin-I. Liao[1]

**1** NTT Communication Science Laboratories, NTT Corporation, Atsugi, Kanagawa, Japan, **2** Department of Computer Science and Engineering, Toyohashi University of Technology, Toyohashi, Aichi, Japan

* suzukiy970@gmail.com

## Abstract

Intrinsically photosensitive retinal ganglion cells (ipRGCs) are known to be sensitive to short-wavelength light (460–480 nm; blue or cyan light) and to play a role in regulating physiological responses such as circadian rhythms. Previous studies have shown that exposure to blue light improves performance on working memory tasks compared with exposure to amber light. However, it remains unclear whether these cognitive benefits via light are attributable to integrated signals across ipRGCs and rod/cone or ipRGC alone. To address this, the present study investigates the specific contribution of ipRGCs to working memory performance using a silent substitution method that selectively manipulates ipRGC activity while minimizing the influence of LMS cone responses. Participants engaged in 1- and 2-back tasks under low- or high-ipRGC activation light, a metameric color perceived as magenta. Results showed that hit rate in the 2-back task was significantly higher under exposure to high-ipRGC light than to low-ipRGC light. Our overall findings provide direct evidence that isolated ipRGC activation, independent of perceptual blue or cone involvement, can modulate cognitive task processing.

## Introduction

Visual processing begins with the detection of light by the photoreceptors in the retina. In addition to rods and cones, which are the primary photoreceptors for light detection, there are intrinsically photosensitive retinal ganglion cells (ipRGCs), which contain melanopsin as a photoreceptor and are maximally sensitive to short-wavelength light (460–480 nm; blue or cyan light). ipRGCs are another type of photoreceptor known to be involved in the pathway of non-image-forming vision [1–4]. It is known that ipRGCs play a role in mediating synchronization of the internal biological clock with the external light–dark cycle (i.e., circadian rhythms) by projecting to the suprachiasmatic nucleus (SCN) and suppressing melatonin secretion in response to light [3,4]. They also contribute to sustained pupil constriction in response to steady blue-rich light through projections to the olivary pretectal nucleus (OPN) [5]

**Data availability statement:** Participants' data and experimental scripts are available from https://github.com/suzuki970/ipRGC-Nback.

**Funding:** This work was supported by Grants-in-Aid for Scientific Research from the Japan Society for the Promotion of Science (grant number 20H05956).

**Competing interests:** The authors have declared that no competing interests exist.

and contribute to enhanced brightness perception in visual processing [6,7]. ipRGC subtypes of M1 cells are primarily involved in non-image-forming functions such as circadian rhythm and pupillary light reflex, while other subtypes of M2–M5 have been implicated in image-forming functions [8,9]. Furthermore, previous studies have reported that exposure to blue light, which includes an ipRGC activation wavelength, improves higher cognitive functions such as vigilance task [10,11] or working memory task [12,13].

When light with a short-wavelength spectrum enters the retina, ipRGCs send signals to multiple brain regions, including subcortical nuclei and cortical areas involved in arousal and cognitive functions [14]. Recent studies have focused on the regulation of circadian rhythms by ipRGC activation, as ipRGC signals project to the SCN, which mediates the circadian clock and synchronizes various parasympathetic and sympathetic nuclei [4]. Furthermore, since the SCN projects to the locus coeruleus (LC), which releases the neurotransmitter norepinephrine (NE), ipRGC activity may be indirectly related to cognitive tasks that require attention, executive functions, and memory, involving the LC-mediated NE release circuit [15]. Based on the observation that blue light involves the LC-mediated pathway, it has been reported that blue light exposure can improve performance on working memory tasks [12,13] as well as reduce sleepiness [16]. On the other hand, another study has failed to find evidence that light can alter neurobehavioral performance on executive function tasks [17]. When comparing blue light with red (or amber, etc.) in the experimental paradigm, however, it is unclear whether these discrepancies are due to the involvement of specific photoreceptors or whether other blue-specific factors, such as color perception related to the autonomic nervous system, are a necessary condition for improvement of task performance [18,19]. Chien et al. (2020) showed that perceived audiovisual synchrony was influenced by the red background rather than blue [20]. Similarly, Yang et al. (2023) found that lighting-induced changes in attention dynamics stemmed from color associations, but not from ipRGC involvement [21]. These findings highlight the importance of isolating the effects of specific photoreceptor to further investigate the function of ipRGCs, to avoid potential confounding effects related to color.

Given the previous findings that ipRGC stimulation via blue light can enhance vigilance and cognitive performance, we hypothesized that selective ipRGC activation—without altering cone responses—would directly improve working memory performance. In this study, we adopted a silent substitution method, which allows selective manipulation of ipRGC activation while keeping the LMS cone responses constant [22–24], to investigate the effect of ipRGC activation on working memory (N-back task). To account for differences in the distribution of cones in each subject, we designed metameric colors for each subject in advance to control the perceived color. We confirmed whether the designed high-ipRGC light was perceived as brighter than the low-ipRGC light before the N-back task. We then investigated whether the task performance on the N-back task would change under exposure to either the low- or high- ipRGC light using a between-subjects design. In addition, we asked for a subjective sleepiness and fatigue rating at several points during the task to investigate whether subjective physiological markers could be affected by the metameric light and explain task performance.

## Method

### Participants

Twenty-seven volunteers (17 men, 10 women; age range: 20–40 years, mean age = 24.77, SD = 5.81) with normal or corrected-to-normal vision and normal color vision (based on participants' verbal reports) participated in the experiments. The experiment was conducted during three time slots: 10:00–13:00, 13:00–16:00 and 16:00–19:00, with 9 participants assigned to each slot. In each time slot, 4, 5, and 4 participants, respectively, began with the low-ipRGC condition. One participant was excluded from the analyses because the number of task responses for both 1- and 2-back tasks was less than three standard deviations away from the mean across participants. The sample size was determined by a priori power analysis using G*Power, considering a power of 0.8, a type I error level of 0.05, and an expected effect size of Cohen's f = 0.3 (corresponding to partial $\eta^2$=0.083) [25]. This effect size was chosen based on the guideline (1988), in which f = 0.3 is considered a medium-to-large effect, providing a reasonable and conservative assumption [26]. These parameter settings led to a minimum sample size of 24 participants. Participants were recruited between May 10, 2024, and June 9, 2024. All experimental procedures were in accordance with the ethical principles outlined by the Research Ethics Committee of Nippon Telegraph and Telephone (NTT) Communication Science Laboratories. All participants provided written informed consent. Participants' data and experimental scripts are available from https://github.com/suzuki970/ipRGC-Nback.

### Lighting system

For the projector system, we utilized two projectors (EH-TW5650, EPSON, Nagano, Japan) to generate metameric pairs of different ipRGC activation level lights, as shown in Fig 1A [24,27]. A diffusion board with dimensions of 505 x 290 mm and a resolution of 1920 x 1080 px was placed 640 mm away from the projectors. The ipRGC primary projector was filtered by a 488 ± 2.0 nm notch filter with a full width at half maximum = 10.00 ± 2.0 (Edmund Optics Inc, NJ, USA, #67–108). Another S-cone primary projector was filtered by a long-pass filter with a cut-on at 435 nm (Edmund Optics Inc, NJ, USA, #66–080). Fig 1B shows the spectrum of maximum output in each RGB channel and projector. The CIE 1931 xy chromaticity coordinates for R, G, and B for the ipRGC primary projector were (0.638, 0.357), (0.358, 0.613), and (0.161, 0.065). Those for the S-cone primary projector were (0.631, 0.366), (0.334, 0.613) and (0.143, 0.138). To generate metameric light that activates ipRGCs at different levels, we used the principles of receptor silent substitution, which allows control of ipRGC levels independently from cone activation (i.e., metameric lights but eliciting different levels of ipRGC activation).

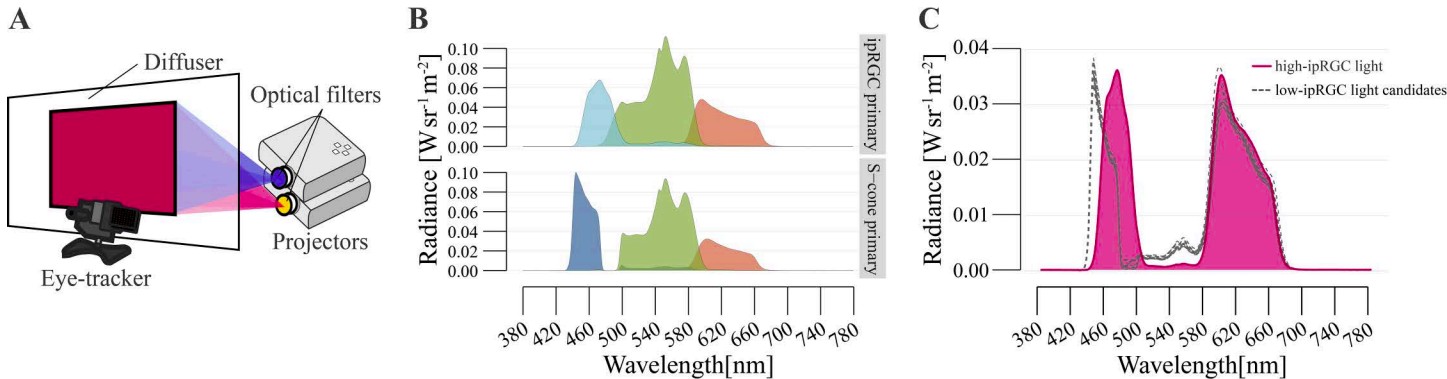

**Fig 1. Experimental design.** (A) Experimental apparatus. Participants seated in front of the diffuser and the stimulus projected from behind by projectors. (B) the maximum output of light spectra for each RGB and projector. ipRGC-primary projector can present cyan light while s-cone projector cut the wavelength at around 480nm. (C) the high-ipRGC light target and low-ipRGC light reference candidates.

We assumed physiologically relevant CIE 2006 photoreceptor sensitivities for the LMS cones [28]. A standardized ipRGC sensitivity function was obtained from the CIE S 026/E:2018 standard.27 using the CIE S 026 Toolbox (v1.49a - November 2020) [29]. The luminance and color of the stimuli were calibrated by a spectroradiometer (CS-3000HDR, KONICA MINOLTA, Tokyo, Japan).

The metameric lights were computationally simulated in advance using the photoreceptors and ipRGC sensitivity model. The CIE xy coordinates for high-ipRGC light were determined to maximize ipRGC activation contrast between the high-ipRGC light and the low-ipRGC light. This results in CIE xy coordinates for the high and low-ipRGC light of (0.412, 0.241) and a luminance of 713.56 cd/m$^2$ for high-ipRGC light which is perceived as magenta. To select metameric low-ipRGC light for individuals, we prepared low-ipRGC light candidates with a slightly different a* and b* color which were selected from a*b* coordinates around the high-ipRGC light at a*$=$189.19, b*$=-$50.04 (central point): one candidate is at the same coordinates as for high-ipRGC light and four candidates are at four equidistant points ($\sqrt{(a^*)^2+(b^*)^2}=1$) from the central point, and seven candidates are at another seven points slightly further away ($\sqrt{(a^*)^2+(b^*)^2}=3$) from the central point, resulting in a luminance range for the low-ipRGC light candidates of 717.58$\pm$20.57 cd/m$^2$. Using these low-ipRGC light candidates, one candidate was selected in the color tuning protocol (see Methods) as the low-ipRGC light for each participant. Fig 1C shows the measured low and high-ipRGC light spectra.

## Stimulus and apparatus

Each participant's head was kept still by a chinrest at a viewing distance of 500 mm from the screen. The task was conducted in a dark room and executed using MATLAB2019a (MathWorks, Natick, MA, USA) and Psychtoolbox [30]. A fixation point in the shape of a cross was located at the center of the screen with a visual angle of 0.243° and the CIE xy coordinates and luminance of the fixation cross were (0.354, 0.391) and 1200 cd/m$^2$.

## Experimental procedure

All participants were seated in front of the computer screen with their heads positioned on a chinrest to prevent unwanted movement. The experiment began with the color tuning and brightness judgment tasks (see details in the following section), as illustrated in the overall procedure in Fig 2A. In the main N-back task, each block was started with a standard five-point eye-gaze calibration. One block consisted of 6 runs, which included 50 trials of 1-back and another 50 trials of 2-back tasks and two questionnaires for the sleepiness and fatigue rating. Sleepiness refers to a feeling of daytime sleepiness. Fatigue refers to fatigue from the n-back task. Participants were instructed to press a button using a keyboard when a target character appeared on the screen according to the task described above. For the questionnaires, participants were asked to rate their subjective sleepiness and fatigue on a scale from 0–10 on a visual analog scale. The participants were able to take a short break while keeping their head on the chinrest between runs. One run lasted approximately 4–5 minutes so the entire block took 24–30 minutes. Exposure to the lighting condition (high- or low–ipRGC stimulation) began precisely at the onset of the initial run of 1-back task and continued throughout the entire block. Participants were not exposed to either lighting condition before the block. There were two blocks and participants were exposed to low and high-ipRGC light during each block, respectively. To reduce any duration effect from light in the first block, considering the long-lasting effect of ipRGC on the brain [31], the blocks were separated by a 60-minute break to washout the effect of the light. Thus, no light stimulation was presented during the inter-block resting period to prevent potential carry-over effects. The light condition was counterbalanced across participants (i.e., 13 participants started with high-ipRGC light in the first block and the other half started with low-ipRGC light).

## Color tuning protocol

We started with a tuning experiment to select the metameric low-ipRGC light for each subject. Participants were asked to answer whether the colors of two presented images were identical or not with a forced choice of "yes" or "no" as shown in

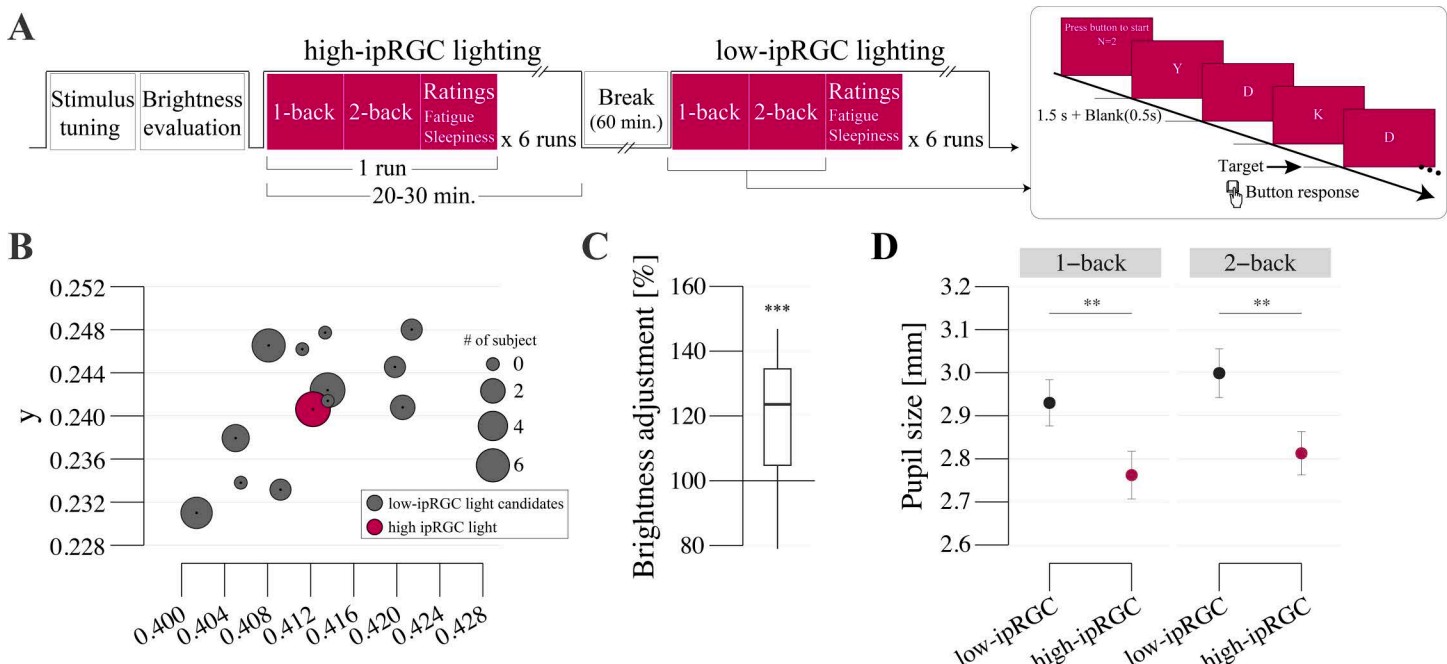

**Fig 2. The pupil size and N-back task accuracy.** (A) Experimental design of stimulus tuning, brightness evaluation, N-back tasks, and subjective ratings. 1- and 2-back tasks and subjective ratings are performed repeatedly within the block using low- or high-ipRGC light (see Method). (B) CIE xy coordinates of low- and high-ipRGC lights. The size of each circle indicates the number of subjects who chose the light as a metamer of high-ipRGC light. (C) Result of brightness adjustment. The vertical line shows the percentage of subjective brightness enhancement of high-ipRGC light compared to low-ipRGC light as selected by participants. (D) Pupil size during low- and high-ipRGC exposure in 1- and 2-back tasks. The asterisks (*) indicate statistical significance at *$p < 0.05$, **$p < 0.01$. Error bars indicate the standard error of the mean.

S1A Fig. To prevent participants from discriminating colors based on differences in brightness of color caused by ipRGC, the color pattern consisted of 4 kinds of circle size and luminance, maintaining the xy coordinates as set out above (see S1A Fig). The visual angle of the stimulus was 9.63°. One trial consisted of a 0.5-second fixation screen followed by a 1-second presentation of the first stimulus. Then, the second stimulus was presented for 1 second, followed by 0.5 seconds of blank screen. Following that, participants responded "yes" or "no" by pressing a button. One of the two presented stimuli was always the high-ipRGC stimulus as a target stimulus. The other stimulus was either a low-ipRGC light chosen from 12 candidates or the same as the target stimulus (i.e., high-ipRGC light) as a reference stimulus. The order of high-ipRGC and low-ipRGC stimuli were randomized. The whole experiment consisted of 13 reference stimuli (12 low-ipRGC candidates + 1 high-ipRGC) × 12 trials = 156 trials, divided into two runs. One run took approximately 7 minutes. The light which was answered as most similar color to high-ipRGC light (i.e., the highest number of answers of "yes") was used as a metameric low-ipRGC light. When two or more lights had the same number of "yes", the light with the closest CIE xy coordinates to the high-ipRGC light was employed as a metameric low-ipRGC light.

**Brightness judgement**

To confirm the activation of ipRGCs, we performed a brightness-matching experiment. The light selected in the stimulus tuning experiment as a reference stimulus was paired with high-ipRGC light as a target stimulus and each was presented at the left or right of a homogeneous screen as illustrated in S1B Fig. The left and right of the screen was divided by a gray vertical band (2.3° in width with 468.1 cd/m²). Participants were asked to adjust the reference stimulus luminance

(i.e., low-ipRGC light) using the up-arrow key to increase and the down-arrow key to decrease its luminance until it was perceived as having identical brightness to the target stimulus (i.e., high-ipRGC light). The color and luminance of high-ipRGC light remained constant during the adjustment period, while only the luminance of the low-ipRGC stimulus changed, with its chromaticity held constant. There were 20 trials in total (approximately 10 minutes), and the reference and target stimulus position (i.e., left or right) was randomized.

### N-back task

A classical alphabet N-back task was performed in a visual modality, as in previous studies, using alphabet characters [31]. The characters 'A', 'B', 'C', 'D', 'E', 'J', 'K', 'O', 'P', 'R', 'S', 'U', 'V', 'X', 'Y', 'Z', which were within the standard deviations from the mean across A-Z, were used in the Arial font and presented within a vertical visual angle of $4.38 \pm 0.05°$. Participants observed a series of these letters randomly presented at the center of the monitor with a white color with CIE xy (0.354, 0.391) and luminance of 1200 $cd/m^2$. One trial consisted of 0.5 seconds of letter presentation and 1.5 seconds of inter-stimulus interval. In the 1-back task, participants were asked to respond using their right hand whether the letter presented in the current trial was identical to the letter presented in the immediately previous trial. In the 2-back task, participants responded as to whether the letter presented in the current trial was identical to the letter presented two trials before. Participants performed the 1-back task first and the 2-back task second within the run. At the beginning of the task, instructions on the kind of task (i.e., 1- or 2-back) were shown on the screen. Then, a fixation cross was presented for 5 seconds.

### Pupillometry and analysis

To confirm whether high-ipRGC light activated participants' ipRGCs, we monitored pupil size, as previous studies have reported that ipRGC activation leads to pupil constriction [5,32]. Pupil size and eye movement during the task were measured using an eye-tracking system (EyeLink 1000, SR Research, Oakland, Canada) at a sampling rate of 1000 Hz. Movement of both eyes was recorded using an infrared light video camera at a resolution of 0.1°. Data from the eye with the least amount of data loss (either the left or right eye) was used for analysis for each subject. The tracking was performed throughout the study based on pupil diameter using the centroid mode. Pupil size was recorded by the eye-tracker in arbitrary units and converted to millimeters from pixels. Pupil data during eye-blinks, which were obtained as zero values in the data, and values that fell outside the 99% confidence interval from the normal distribution of the whole experiment for each participant, were interpolated by cubic Hermite interpolation.

### Statistics

A two-way repeated-measures analysis of variance (ANOVA) was performed using the light condition and N-back as within-subject factors. The level of statistical significance was set to $p < 0.05$ for all analyses. Pairwise comparisons of the main effects were corrected through multiple comparisons using the modified Bonferroni method. Effect sizes were given as partial $\eta^2$; $\eta_p^2$ for ANOVA and as d; $d_z$ for t-tests. To quantify the evidence in the data, we performed Bayesian one-sample t-tests using the BayesFactor package (v0.9.12-4.2) [33] for the R software (Version 3.6.3) [34]. We report the Bayesian Factor (BF), estimating the relative weight of the evidence in favor of $H_1$ over $H_0$ as $BF_{10}$. Greenhouse-Geisser corrections were performed when the results of Mauchly's sphericity test were significant. To model the rating scores and RTs using light conditions for each participant, we used a linear mixed-effects modeling with participants as a random effect to fit the data using the lme4 packages [35].

## Results

### Stimulus tuning experiment

In order to tune a metameric stimulus that has a higher ipRGC contrast but is independent of cone activities taking into account individual differences in cone distribution (reference), we first identified a metameric light with high-ipRGC light

by a 2AFC experiment. Fig 2B shows the CIE xy coordinates of the candidate low-ipRGC lights and the number of participants for whom each light was judged to be regarded as a metameric match with the high-ipRGC light. In all subjects, the selected light was reported as being perceived as the same color at least once (S1C Fig). The averaged luminance of selected low-ipRGC light calculated based on CIE photoreceptor sensitivities was $708.615 \pm 17.355$ cd/m². The average Michelson contrast for ipRGC across the used metamer pair was $24.757 \pm 2.517$.

## Brightness evaluation

Using the metameric light for each participant, we confirmed the activation of ipRGC by a brightness judgement experiment. Fig 2C shows the ratio of adjusted luminance to high-ipRGC light. The high-ipRGC light was perceived as around 20% brighter than the low-ipRGC light ($t(25) = 4.943$, $p < 0.001$, $d_z = 1.371$, $BF_{10} > 100$), consistent with previous studies [6,7].

## Pupil size

Pupil size was measured using an infrared camera while participants were presented with low and high-ipRGC light and engaged in the 1- and 2-back tasks (see Method). Fig 2D illustrates average pupil size during the 1- and 2-back tasks in each lighting condition. A two-way repeated measures ANOVA for the main effect of light showed that the pupil size for high-ipRGC light was significantly smaller than for low-ipRGC light ($F(1,25) = 21.614$, $p < 0.001$, $\eta_p^2 = 0.464$, $BF_{10} > 100$), consistent with previous studies as an effect of activation of ipRGC. Post analysis also showed that the pupil size in both N-back tasks for high-ipRGC light was smaller than for low-ipRGC light for both 1- and 2-back tasks (1-back: $t(25) = -4.489$, $p < 0.001$, $d_z = -0.603$, $BF_{10} > 100$; 2-back: $t(25) = -4.619$, $p < 0.001$, $d_z = -0.676$, $BF_{10} > 100$, respectively). In addition, pupil size during the 2-back task was larger than that during the 1-back task, an effect of cognitive load differences ($F(1,25) = 14.086$, $p = 0.001$, $\eta_p^2 = 0.36$, $BF_{10} = 1.38$), as reported here and previously elsewhere [36].

## Accuracy

To see the effect of light on the N-back task performance, we compared the hit rate between the lighting conditions, as illustrated in Fig 3A. We found a main effect of light showing that the hit rate under high-ipRGC light was higher than under low-ipRGC light ($F(1,25) = 4.872$, $p = 0.037$, $\eta_p^2 = 0.163$, $BF_{10} = 0.675$). There is significant interaction between N-back and light condition ($F(1,25) = 5.298$, $p = 0.03$, $\eta_p^2 = 0.175$, $BF_{10} = 0.654$). Post analysis showed that the higher hit rate under high-ipRGC light was seen in the 2-back but not in the 1-back tasks (1-back: $t(25) = 0.702$, $p = 0.489$, $d_z = 0.078$, $BF_{10} = 0.259$, 2-back: $t(25) = 2.728$, $p = 0.011$, $d_z = 0.508$, $BF_{10} = 4.2$, respectively). As an effect of working memory load on the accuracy, the hit rate for the 1-back task is higher than that for the 2-back task ($F(1,25) = 30.83$, $p < 0.001$, $\eta_p^2 = 0.552$, $BF_{10} > 100$). The number of false alarms also decreased in the 2-back task under high-ipRGC light with marginal significance ($t(25) = 1.914$, $p = 0.067$, $d_z = 0.421$, $BF_{10} = 1.003$). The details of hit, miss, false alarm and correct rejection are shown in Table S1.

## Response times

Response times (RTs) were calculated from the stimulus onset to the participants' key presses. Linear mixed model analyses were conducted corresponding to the following formula: $RTs \sim Light + (1|Subject)$. To statistically assess whether the RTs differed between lighting conditions, 95% confidence intervals (CI) for the RTs were estimated with 10000 bootstrap samples. We found the RTs were faster under the high-ipRGC light than under the low-ipRGC light for 1-back (CI[−28.25, −9.044], $t(25) = 4.536$, $p < 0.001$, $d_z = 0.139$) but not for 2-back tasks (CI[−18.744, 10.331], $t(25) = 0.364$, $p = 0.716$, $d_z = 0.011$) as shown in Fig 3B.

## Questionnaire

To test whether the light condition affected participants' sleepiness and fatigue, we asked participants to rate their subjective sleepiness and fatigue on a scale of 0–10. The fixed effects of light, run, and their interaction model including block

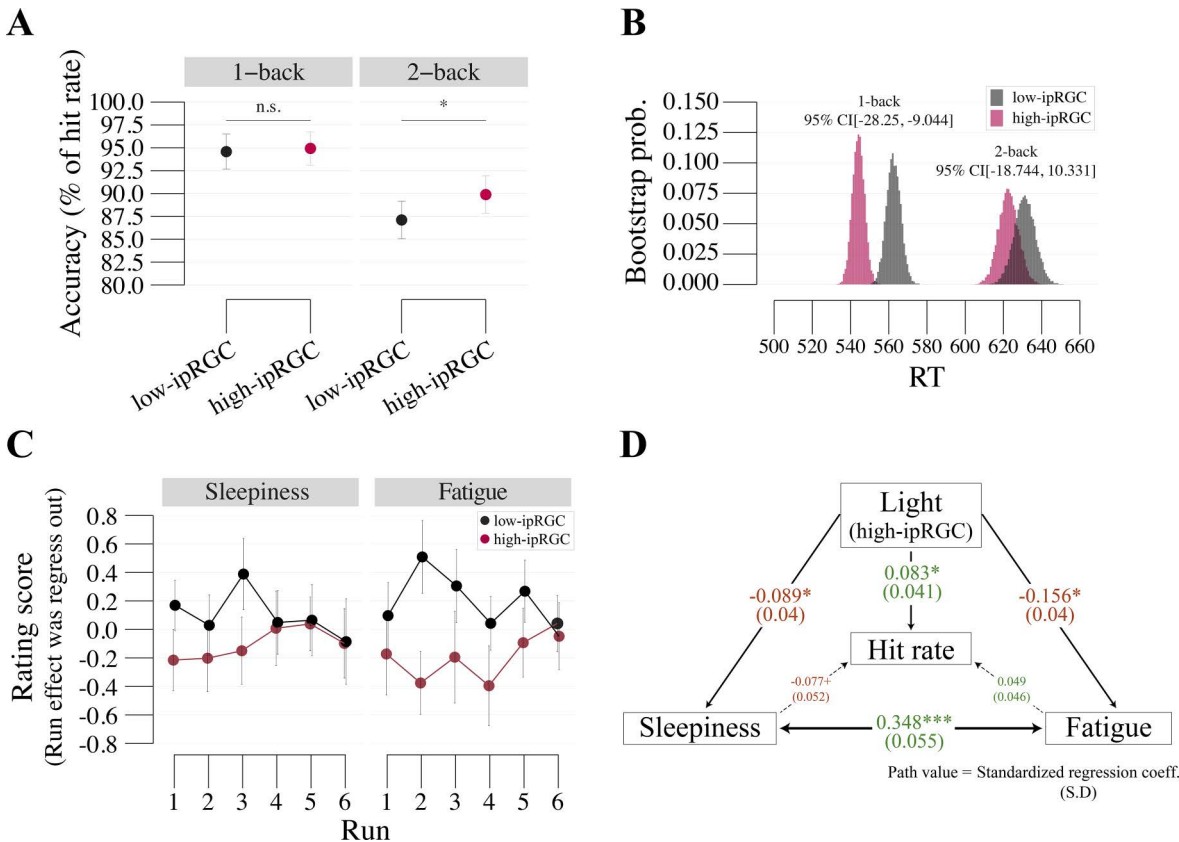

**Fig 3. Subjective rating and model weight.** (A) Accuracy (hit rate for 1- and 2-back tasks) in each light condition. (B) The bootstrapped RT was significantly faster under the high-ipRGC light than under the low-ipRGC light. (C) Results for subjective ratings of sleepiness and fatigue in each run and light condition. (D) Modeling of the Relationship between light condition, hit rate, and sleepiness using mediation analysis. Path values indicate standardized regression coefficients with standard errors, estimated with 10000 bootstrap samples. + p < 0.1, *p < 0.05, ***p < 0.001. Error bars indicate the standard error of the mean.

order are reported here; we fitted the rating scores with light, run, and block order as within-subject factors and subject as a between-subject factor. The block order refers to whether the high-ipRGC light block or low-ipRGC light block was performed in the first block. The scores were negatively affected by the light condition, indicating that the high-ipRGC light contributed to a lower subjective score for both sleepiness and fatigue (sleepiness: $t(25) = -2.159$, $p=0.032$, $d_z = 0.263$; fatigue: $t(25) = -4.602$, $p < 0.001$, $d_z = 0.562$).

We fitted the rating scores with run and participants as a random effect using a linear mixed model, showing that the subjective sleepiness and fatigue scores increased significantly with an increasing effect of time-on-task (sleepiness: $t(25) = 8.895$, $p < 0.001$, $d_z = 1.079$; fatigue: $t(25) = 8.709$, $p < 0.001$, $d_z = 1.057$). Using the coefficient of the model, we regressed out the time-on-task effect from the rating scores as shown in Fig 3C.

To examine whether pupil size could be attributed to variations in subjective fatigue and sleepiness, we computed correlations between pupil size and these subjective ratings. As shown in S2 Fig, we observed a significant correlation between pupil size and fatigue ratings, but not with sleepiness. Additionally, we confirmed that the degree of pupil constriction under high-ipRGC light was not predictive of the individual brightness adjustment values in the brightness-matching experiment.

To determine how the light condition, the hit rate, sleepiness, and fatigue affected each other, we performed a mediation analysis as shown in Fig 3D. We found the light condition significantly affected hit rate, sleepiness, and fatigue (hit rate: $\beta$ = 0.083, $p$ = 0.044, $d_z$ = 0.165; sleepiness: $\beta$ = −0.089, $p$ = 0.029, $d_z$ = 0.179; fatigue: $\beta$ = −0.156, p < 0.001, $d_z$ = 0.316). Although sleepiness negatively affected the hit rate with a weak effect (i.e., lower sleepiness improved the hit rate), we did not find strong evidence that the sleepiness and fatigue rating can explain the hit rate (sleepiness: $\beta$ = −0.077, $p$ = 0.075, $d_z$ = 0.145; fatigue: $\beta$ = 0.049, $p$ = 0.266, $d_z$ = 0.091).

## Discussion

The present study sought to ascertain whether the ipRGC-related circuit plays a role in working memory task performance (N-back task) using a silent substitution method with a projector and optical filters [24,27] to minimize the involvement of other photoreceptors with varying ipRGC activation levels. Participants were exposed to metameric lights during the 1- and 2-back tasks in successive different blocks. Consistent with previous studies, the high-ipRGC light significantly constricted pupil size [5,32], and enhanced brightness perception [6,7]. The results demonstrated that the hit rate in the 2-back task was significantly higher under exposure to the high-ipRGC light than under exposure to the low-ipRGC light. Furthermore, high-ipRGC light can reduce subjective sleepiness and fatigue. These findings suggest that ipRGC activation contributes to visual input and working memory integration.

Previous studies have reported that working memory task performance, measured by reaction time and hit rate, was enhanced by blue light exposure compared to red or green light [31]. Blue light exposure has been demonstrated to elevate activity in the left dorsolateral prefrontal cortex (DLPFC) and the right ventrolateral prefrontal cortex [31,37], or both left and right DLPFC [38], which is presumed to be conveyed from the ipRGC signal to a neural circuit that involves a process of working memory task. A recent study on the selective activation of ipRGCs found that the frontal eye field, which is involved in higher cognitive functions, is activated by high-ipRGC metameric light [39]. In addition, a study using ipRGC knockout mice demonstrated that light activates the prefrontal cortex via an ipRGC–thalamic–corticolimbic pathway, highlighting a role of ipRGCs in cortical regulation [40]. Our findings also support this idea in that the improved working memory performance under high-ipRGC light suggests that PFC activity may facilitate the integration of cortical networks and executive control. Recent evidence suggests that lighting conditions (e.g., bright versus dim) influence memory by modulating retinal ganglion cell (RGC) signals that are relayed to the hippocampus via projections from the ventral lateral geniculate nucleus and intergeniculate leaflet (vLGN/IGL) [41]. In a mouse study, bright light disrupted object recognition memory, whereas this effect was absent in melanopsin-deficient mice, indicating that both melanopsin-driven and rod/cone-driven photoreceptor responses are integrated to mediate the effect of light on memory performance [42]. Thus, it is essential to control for the stimulation of these photoreceptors across experimental conditions to isolate the role of ipRGCs in cognitive control [43]. Similarly, because comparisons between blue light and short- or medium-wavelength lights inherently involve differential stimulation of cone and rod, color differences influence pathways projecting to cortical areas, potentially affecting arousal, emotion, relaxation, and heart rate [18,19]. This issue—namely, psychological and physiological effects elicited by light and color—was addressed in the current study by utilizing metameric lights perceived as magenta, thereby excluding the potential influence of color perception. Therefore, it is noteworthy that our lighting condition yielded enhanced task performance and diminished subjective feelings of sleepiness and fatigue when participants were exposed to light that strongly activates ipRGCs.

The high-ipRGC light exposure reduced subjective sleepiness and fatigue, which aligns with previous research indicating less subjective sleepiness with 6500K light (which exhibits heightened ipRGC activation) in comparison to pre- and post-illumination lighting [16], and bright light exposure has been shown to reduce sleepiness levels in comparison to dim light [44,45]. A direct neural projection exists from ipRGCs to the SCN, which is either directly or indirectly connected to brain regions associated with arousal levels. It can be postulated that light that activates ipRGCs may increase or prevent a decrease in arousal levels by suppressing melatonin secretion by the pineal gland. Related to changes in arousal

level, the increase in hit rate observed under high-ipRGC light in this study was expected to be mediated by psychological perceptions such as sleepiness. However, while less sleepiness partially contributed to a higher hit rate, our mediation analysis did not provide strong evidence that sleepiness and fatigue could explain this improvement in performance. Taken together, our overall results highlight the potential effect of ipRGC activation on hit rate, sleepiness, and fatigue, while psychological factors, specifically mediated by ipRGC modulation, may have limited effect on the hit rate.

The findings revealed that the impact of an elevated task performance under high-ipRGC light exposure was confined to the 2-back task and not observed in the 1-back task. On the other hand, faster reaction times were observed in the 1-back task but did not reach the level of significance in the 2-back task. As expected, the working memory load for the 1-back task was expected to be lower, as evidenced by the observation of smaller pupil sizes during the task (Fig 2D). Working memory load increases as n increases, and so does the activation intensity of the lateral prefrontal cortex (LPFC) [46,47]. This may indicate that ipRGC activation affects cognitive behavioral performance exclusively during executive tasks when the DLPFC activation intensity is relatively higher, as in at least N≧2. Notably, we failed to find differences in reaction times between the ipRGC lighting conditions for the 2-back task. This finding seems consistent with previous research findings that LPFC activity was not related to reaction times in between-subject variability [47]. Additionally, given that the experimental duration was relatively long in comparison to previous studies, it is possible that participants might have prioritized performing the task correctly over responding quickly when performing the 2-back task.

Our findings provide direct evidence that ipRGC play a modulatory role in cognitive task processing, independent of perceptual blue or cone involvement. These results align with the ipRGC role in projecting to brain areas involved in working memory tasks. The application of light that modulates ipRGC activation without involving the user's perceptual color experience could enhance cognitive performance independent of blue light perception, broadening potential applications for lighting environments to support cognitive function.

## Supporting information

**S1 Fig. Experimental design for stimulus tuning and brightness experiment.** (A) Experimental design for stimulus tuning and (B) brightness evaluation (C) The response probability of "identical" in the stimulus tuning experiment. The bright circles are chosen as metamer light as low ipRGC condition for each subject as illustrated in different colors.
(PDF)

**S2 Fig. Correlations between pupil size and fatigue, sleepiness, and brightness adjustment.** lot showing the relationship between pupil size and sleepiness ratings. (B) Scatter plot showing the relationship between pupil size and fatigue ratings. In both (A) and (B), circle markers represent data from the low-ipRGC light condition, and triangle markers represent data from the high-ipRGC light condition. (C) Scatter plot showing the relationship between the degree of pupil constriction under high-ipRGC light (horizontal axis) and the brightness adjustment values from the brightness-matching experiment (vertical axis). Each color represents data from an individual participant.
(PDF)

**S1 Table. Hit, miss, false alarm (FA) and correct rejection (CR) rate in 1- and 2-back task.**
(DOCX)

## Author contributions

**Conceptualization:** Yuta Suzuki, Shigeki Nakauchi, Hsin-I. Liao.

**Data curation:** Yuta Suzuki.

**Formal analysis:** Yuta Suzuki.

**Funding acquisition:** Shigeki Nakauchi.

**Investigation:** Yuta Suzuki.

**Methodology:** Yuta Suzuki.

**Project administration:** Yuta Suzuki.

**Software:** Yuta Suzuki.

**Supervision:** Hsin-I. Liao.

**Validation:** Yuta Suzuki.

**Visualization:** Yuta Suzuki.

**Writing – original draft:** Yuta Suzuki.

**Writing – review & editing:** Yuta Suzuki, Shigeki Nakauchi, Hsin-I Liao.

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
