## [Decision Letter · Decision Letter 0]

Dear Dr. Suzuki,

Thank you for submitting your manuscript to PLOS ONE. After careful consideration, we feel that it has merit but does not fully meet PLOS ONE’s publication criteria as it currently stands. Therefore, we invite you to submit a revised version of the manuscript that addresses the points raised during the review process.

We look forward to receiving your revised manuscript.

Kind regards,

Steven Barnes

Academic Editor

PLOS ONE

 [This work was supported by Grants-in-Aid for Scientific Research from the Japan Society for the Promotion of Science (grant number 20H05956)]. 

[This work was supported by Grants-in-Aid for Scientific Research from the Japan Society for the Promotion of Science (grant number 20H05956)]

 [This work was supported by Grants-in-Aid for Scientific Research from the Japan Society for the Promotion of Science (grant number 20H05956)]. 

4. We notice that your supplementary figures are uploaded with the file type 'Figure'. Please amend the file type to 'Supporting Information'. Please ensure that each Supporting Information file has a legend listed in the manuscript after the references list.

Additional Editor Comments:

We apologize that this has taken so long, but we were only able to recruit one referee for assessment of your manuscript after sending out many invitations. The sole reviewer provided a well-balanced assessment, and we invite you to respond and fully address the referee's concerns.

Reviewers' comments:

Reviewer's Responses to Questions

**Comments to the Author**

1. Is the manuscript technically sound, and do the data support the conclusions?

Reviewer #1: Yes

2. Has the statistical analysis been performed appropriately and rigorously?

Reviewer #1: Yes

3. Have the authors made all data underlying the findings in their manuscript fully available?

Reviewer #1: Yes

4. Is the manuscript presented in an intelligible fashion and written in standard English?

Reviewer #1: Yes

Reviewer #1: Review for PONE-D-25-05696

This paper discussed the effects of high- versus low-ipRGCs stimulations on working memory performance. They revealed a higher hit rate and faster reaction time under higher versus low-ipRGCs condition in a 2-back and 1-back working memory task, respectively. The experimental manipulations were well-controlled, and the manuscript was well-written with clear rationale and information provided. Here are some suggestions that I would recommend the authors to add to the manuscript to make the manuscript even clearer.

Abstract

1. The authors could make the gap in previous research clearer. Specifically, previous studies did not control the light conditions well and usually led to mixed results, which motivated this study to examine the n-back effect under high vs. low ipRGCs conditions.

Introduction

1. Line 37, the authors an specify the direction of how the circadian rhythms and pupil constrictions are influenced by ipRGCs.

2. Line 50, Hung et al. (2017) also proposed neural circuits that can mediate the role of ipRGCs in higher cognitive functions.

3. Line 52, what does it mean to decrease physiological effects such as sleepiness?

4. Line 54–56: There could be several reasons why different background lighting conditions lead to different conclusions. For example, using silent substitution, Chien et al. (2020) demonstrated that the perceived audiovisual simultaneity difference did not originate from the influence of ipRGCs, but rather from the red background. Using a similar approach, Yang et al. (2023) compared the temporal dynamics of endogenous and exogenous attention shifting under different lighting conditions. However, they concluded that ipRGCs were not involved in these processes; instead, the effects were attributed to stereotypical associations with physical light colors. The authors could highlight more such cases to emphasize the importance of carefully inferring the effects of ipRGCs

Methods

1. The authors should specify the timing when the subjects conducted the task (e.g., morning, afternoon, evening or so).

2. How long did the subjects expose to ipRGCs-high and ipRGCs-low light before the task started? This is important given that ipRGCs are sluggish in their activity (Wong, 2012).

3. Line 79, why did the authors expect the effect size to be 0.3?

4. Figure S1A, the timing (e.g., 1s, 0.5s) can be placed at the second row, otherwise it is too crowded in the figure now.

5. Figure S1B, were the high and low ipRGCs patches flashing while the subjects were doing the brightness evaluation? Such as the flicker photometry approach (Bone & Landrum, 2004; Lee et al., 1988).

6. It is not clear to me why the authors analyzed the pupil size for ipRGCs-high and ipRGCs-low conditions, but the brightness levels were different in these two conditions. Was the pupil size correlated with the sleepiness or fatigue ratings? The motivation needs to be clearer here.

7. The data curation is very clear and thorough on Github. I really appreciate it.

Results

1. Why didn’t the authors analyze the dprime but the hit rate here?

2. Line 282, the degree of freedom is 26, but according to the methods, the sample size was n=27-1, please confirm.

3. Please provide the effect size (d value) for the rest of the t-tests.

Discussion

1. The authors proposed several possible neural mechanisms for why ipRGCs are enhancing the working memory performance. However, several of them did not have good control conditions either. I suggest the authors take a look at some latest review regarding how ipRGCs are influencing cognitive functions (e.g., Mahoney & Schmidt, 2024; Meng et al., 2025).

References:

Bone, R. A., & Landrum, J. T. (2004). Heterochromatic flicker photometry. Archives of Biochemistry and Biophysics, 430(2), 137-142.

Chien, S.-E., Chen, Y.-C., Matsumoto, A., Yamashita, W., Shih, K.-T., Tsujimura, S.-i., & Yeh, S.-L. (2020). The modulation of background color on perceiving audiovisual simultaneity. Vision Research, 172, 1-10.

Hung, S.-M., Milea, D., Rukmini, A. V., Najjar, R. P., Tan, J. H., Viénot, F., Dubail, M., Tow, S. L. C., Aung, T., & Gooley, J. J. (2017). Cerebral neural correlates of differential melanopic photic stimulation in humans. Neuroimage, 146, 763-769.

Lee, B., Martin, P., & Valberg, A. (1988). The physiological basis of heterochromatic flicker photometry demonstrated in the ganglion cells of the macaque retina. The Journal of Physiology, 404(1), 323-347.

Mahoney, H. L., & Schmidt, T. M. (2024). The cognitive impact of light: illuminating ipRGC circuit mechanisms. Nature Reviews Neuroscience, 25(3), 159-175.

Meng, J., Huang, X., Ren, C., & Xue, T. (2025). Non-image-forming functions of Intrinsically Photosensitive Retinal Ganglion Cells. Annual Review of Neuroscience, 48.

Wong, K. Y. (2012). A retinal ganglion cell that can signal irradiance continuously for 10 hours. Journal of Neuroscience, 32(33), 11478–11485.

Yang, C.-C., Tsujimura, S.-i., & Yeh, S.-L. (2023). Blue-light background impairs visual exogenous attention shift. Scientific Reports, 13(1), 3794.

**Do you want your identity to be public for this peer review?** For information about this choice, including consent withdrawal, please see our Privacy Policy

Reviewer #1: **Yes: ** Hsing-Hao Lee

---

## [Author Response · Author response to Decision Letter 1]

6 Jun 2025

Reviewer 1’s comments:

Point 1.

This paper discussed the effects of high- versus low-ipRGCs stimulations on working memory performance. They revealed a higher hit rate and faster reaction time under higher versus low-ipRGCs condition in a 2-back and 1-back working memory task, respectively. The experimental manipulations were well-controlled, and the manuscript was well-written with clear rationale and information provided. Here are some suggestions that I would recommend the authors to add to the manuscript to make the manuscript even clearer.

We sincerely appreciate your positive evaluation of our study. We welcome your suggestions for further improving the clarity of the manuscript, and we have carefully addressed each of them in detail below. We believe the revisions have significantly improved the clarity and quality of the manuscript.

Point 2.

Abstract

1. The authors could make the gap in previous research clearer. Specifically, previous studies did not control the light conditions well and usually led to mixed results, which motivated this study to examine the n-back effect under high vs. low ipRGCs conditions.

Thank you for the suggestion. We have clarified the gap between previous research and current study in the Abstract. Specifically, we now emphasize our motivation to isolate the role of ipRGCs in modulating working memory performance, given the confounding influence of rod/cone photoreceptors in prior studies.

Point 3.

Introduction

1. Line 37, the authors an specify the direction of how the circadian rhythms and pupil constrictions are influenced by ipRGCs.

Thank you for the comment. We revised Line 38-43 to specify that ipRGCs mediates a synchronization of circadian rhythms via projections to the suprachiasmatic nucleus (SCN), and contribute to sustained pupil constriction through activation of the olivary pretectal nucleus (OPN).

Point 4.

2. Line 50, Hung et al. (2017) also proposed neural circuits that can mediate the role of ipRGCs in higher cognitive functions.

Thank you for the suggestion. We have added a citation to Hung et al. (2017) in Line 70-74 and described their proposed neural circuits to further support our idea of ipRGC involvement in higher cognitive functions.

Point 5.

3. Line 52, what does it mean to decrease physiological effects such as sleepiness?

Thank you for pointing this out. We clarified the sentence to state more precisely: “blue light exposure can reduce sleepiness.” (in Line 57-59)

Point 6.

4. Line 54–56: There could be several reasons why different background lighting conditions lead to different conclusions. For example, using silent substitution, Chien et al. (2020) demonstrated that the perceived audiovisual simultaneity difference did not originate from the influence of ipRGCs, but rather from the red background. Using a similar approach, Yang et al. (2023) compared the temporal dynamics of endogenous and exogenous attention shifting under different lighting conditions. However, they concluded that ipRGCs were not involved in these processes; instead, the effects were attributed to stereotypical associations with physical light colors. The authors could highlight more such cases to emphasize the importance of carefully inferring the effects of ipRGCs

Thank you for the insightful comment. We added the examples of Chien et al. (2020) and Yang et al. (2023) in Lines 64-69 in the Introduction, highlighting prior studies using silent substitution methods have shown cognitive effects that were driven by background lighting color rather than ipRGC activation.

Point 7.

Methods

1. The authors should specify the timing when the subjects conducted the task (e.g., morning, afternoon, evening or so).

Thank you for your suggestion. The subjects conducted the task from 10 a.m. to 7 p.m., with the subject number counterbalanced and equally distributed in each time slot. We have added the detailed description in the Methods section (under “Participants”), Lines 90-93.

Point 8.

2. How long did the subjects expose to ipRGCs-high and ipRGCs-low light before the task started? This is important given that ipRGCs are sluggish in their activity (Wong, 2012).

Participants were exposed to either high- or low-ipRGC light, which began simultaneously with the onset of the initial n-back task and continued throughout the entire duration of the block (i.e., 20-30 minutes). There was no light stimulation before the task started. During the rest periods between blocks, no light stimulation from either condition was presented. We have now clarified this in the Experimental procedure section to ensure readers understand the timing of light exposure in relation to task onset.

Point 9.

3. Line 79, why did the authors expect the effect size to be 0.3?

We chose an effect size of 0.3 based on two considerations. Our study employed a within-subjects 2 (light condition: high- vs. low-ipRGCs) × 2 (task type: 1-back vs. 2-back) repeated-measures ANOVA, and prior studies using similar paradigms have reported moderate effect sizes. This value aligns with Cohen’s guideline (1988), in which an effect size (partial η²) of 0.3 is considered moderate. Given the exploratory nature of our study and the limited prior data specific to ipRGC stimulation and working memory, we considered this a reasonable and conservative assumption. We have clarified this rationale in the manuscript (Line 97-98).

Point 10.

4. Figure S1A, the timing (e.g., 1s, 0.5s) can be placed at the second row, otherwise it is too crowded in the figure now.

We appreciate your suggestion. We have revised FigureS1A by relocating the timing labels (e.g., 1s, 0.5s) in the second row.

Point 11.

5. Figure S1B, were the high and low ipRGCs patches flashing while the subjects were doing the brightness evaluation? Such as the flicker photometry approach (Bone & Landrum, 2004; Lee et al., 1988).

We did not use a flicker photometry in this experiment because ipRGC-mediated responses are known to be slow (up to approximately 5 Hz) and temporally sustained (e.g., Joyce et al.,. 2016; Zele et al 2018,). Given that flicker photometry paradigms typically rely on fast temporal resolution, we expected that the method would not be well suited for targeting ipRGC-driven brightness perception. Thus, in our brightness matching task, high- and low-ipRGC light patches were presented statically on the left and right sides of the screen. Participants adjusted the luminance of the low-ipRGC patch using “up” and “down” keys until it was perceived to match the brightness of the high-ipRGC patch. We have revised the description in the Brightness judgement secsion to clarify this procedure in greater detail.

Point 12.

6. It is not clear to me why the authors analyzed the pupil size for ipRGCs-high and ipRGCs-low conditions, but the brightness levels were different in these two conditions. Was the pupil size correlated with the sleepiness or fatigue ratings? The motivation needs to be clearer here.

We analyzed the pupil size, as a physiological marker of ipRGC activation, since previous studies have reported that ipRGC activation leads to pupil constriction (e.g., Zele et al., 2019). We have clarified this motivation in the Pupillometry and Analysis section of the manuscript. As originally stated in the manuscript, the brightness matching judgment was also meant to confirm the ipRGC activation. The pupil size was found to correlate with the sleepiness ratings, but not fatigue or brightness. We have added these analyses and results in Questionnaire seccion in Result part and to the Supplementary Information as Figure S2.

Point 13.

7. The data curation is very clear and thorough on Github. I really appreciate it.

Thank you very much for your kind comment.

Point 14.

Results

1. Why didn’t the authors analyze the dprime but the hit rate here?

We opted not to use d-prime since in our data, particularly for the 1-back condition, we observed that several participants had hit rates of 1 (i.e, 100% correct) or false alarm rates of 0. These values lead to unstable or undefined z-scores, potentially distorting d-prime estimates, resulting in inflated or distorted estimates of sensitivity (Macmillan & Creelman, 2005). To avoid these issues and maintain robust comparability across participants, we chose to report hit rate as a more stable and interpretable measure of performance in this context.

Point 15.

2. Line 282, the degree of freedom is 26, but according to the methods, the sample size was n=27-1, please confirm.

Thank you for pointing out the discrepancy regarding the degrees of freedom. In addition to the point, we noticed that, for the brightness judgment, we mistakenly included all the participants’ data for the analysis in the previous version. We have now corrected these results in the revised manuscript.

Point 16.

3. Please provide the effect size (d value) for the rest of the t-tests.

Thank you for pointing this out. We have now added the effect size (Cohen’s d) for all reported statistical tests throughout the manuscript.

Point 17.

Discussion

1. The authors proposed several possible neural mechanisms for why ipRGCs are enhancing the working memory performance. However, several of them did not have good control conditions either. I suggest the authors take a look at some latest review regarding how ipRGCs are influencing cognitive functions (e.g., Mahoney & Schmidt, 2024; Meng et al., 2025).

Thank you for your helpful comment. We have added recent findings reviewed in Mahoney & Schmidt (2024) and Meng et al. (2025) into the second paragraph of the Discussion section. We strengthened our argument for the importance of controlling the influence of other photoreceptors in order to isolate the specific effects of ipRGCs on cognitive control. We believe that these additions help contextualize our experimental designs that selectively activates ipRGC.

---

## [Decision Letter · Decision Letter 1]

Dear Dr. Suzuki,

Thank you for submitting your manuscript to PLOS ONE. After careful consideration, we feel that it has merit but does not fully meet PLOS ONE’s publication criteria as it currently stands. Therefore, we invite you to submit a revised version of the manuscript that addresses the points raised during the review process.

There remain two minor suggestions from the referee for improving the manuscript. Your response to these will be accessed by this Academic Editor alone.  

We look forward to receiving your revised manuscript.

Kind regards,

Steven Barnes

Academic Editor

PLOS ONE

Journal Requirements:

Additional Editor Comments :

The referee responded to your corrections quickly with two, minor comments that need to be addressed. Please return the manuscript to me with your responses.

Reviewers' comments:

Reviewer's Responses to Questions

**Comments to the Author**

Reviewer #1: All comments have been addressed

2. Is the manuscript technically sound, and do the data support the conclusions?

Reviewer #1: Yes

3. Has the statistical analysis been performed appropriately and rigorously?

Reviewer #1: Yes

4. Have the authors made all data underlying the findings in their manuscript fully available?

Reviewer #1: Yes

5. Is the manuscript presented in an intelligible fashion and written in standard English?

Reviewer #1: Yes

Reviewer #1: I appreciate the authors for their work on the revised manuscript. I think this manuscript is almost ready to be published. I have two remaining points that I hope the authors can address before publishing this article.

1. Line 82, reference [22] and that sentence came out of nowhere. I will suggest the authors put this reference in the discussion section.

2. From my understanding, partial eta-squared =0.14 is already a large effect size, I am not sure where and how the authors derived 0.3 still.

With these being said, I am sure that the authors can address these two minor points quickly.

**Do you want your identity to be public for this peer review?** For information about this choice, including consent withdrawal, please see our Privacy Policy

Reviewer #1: **Yes: ** Hsing-Hao Lee

---

## [Author Response · Author response to Decision Letter 2]

13 Jun 2025

Reviewer 1’s comments:

Reviewer #1: I appreciate the authors for their work on the revised manuscript. I think this manuscript is almost ready to be published. I have two remaining points that I hope the authors can address before publishing this article.

We sincerely thank the reviewer for further feedback on our manuscript. Below, we address your two remaining points.

Point 1. Line 82, reference [22] and that sentence came out of nowhere. I will suggest the authors put this reference in the discussion section.

Thank you for your helpful suggestion. We have moved the sentence and reference to the Discussion section (Line 426-428), where it fits within the context of the content.

Point 2. From my understanding, partial eta-squared =0.14 is already a large effect size, I am not sure where and how the authors derived 0.3 still.

Thank you for pointing this out. You are absolutely right. To clarify, the effect size value of 0.3 we referred to during our power analysis is Cohen’s f, not partial eta-squared. We apologize for the misstatement in our earlier version. We have now corrected the manuscript to clearly state that we used an expected effect size of Cohen’s f = 0.3, which corresponds to η_p^2=0.083.

---

## [Editor Report · Decision Letter 2]

Selective activation of ipRGC modulates working memory performance

PONE-D-25-05696R2

Dear Dr. Suzuki,

We’re pleased to inform you that your manuscript has been judged scientifically suitable for publication and will be formally accepted for publication once it meets all outstanding technical requirements.

Kind regards,

Steven Barnes

Academic Editor

PLOS ONE

Additional Editor Comments (optional):

Thank you for your rapid responses to the Reviewer's concerns.

---

## [Editor Report · Acceptance letter]

PONE-D-25-05696R2

PLOS ONE

Dear Dr. Suzuki,

I'm pleased to inform you that your manuscript has been deemed suitable for publication in PLOS ONE. Congratulations! Your manuscript is now being handed over to our production team.

Kind regards,

on behalf of

Dr. Steven Barnes

Academic Editor

PLOS ONE